## Research Article

blue economy; coastal tourism; coral reefs; ecotourism; restoration

**Corresponding author:**
Sierra Garcia;
Email: sierra.garcia@fulbrightmail.org

# Seeing the forest for the coral trees: involvement and perceptions of reef threats among coral restoration volunteers in Roatan, Honduras

Sierra Garcia[1] and Antonella Rivera[2]

[1]The Fulbright Program, West End, Honduras and [2]Coral Reef Alliance, West End, Honduras

## Abstract

Despite the global expansion of coral restoration initiatives that depend on volunteer divers in supporting these programs, research exploring their motivations, sentiments and knowledge remains scarce. This study employed a mixed-methods analysis of surveys ($n = 83$) and interviews ($n = 15$) of a heterogeneous population of coral restoration volunteers and a control group of divers in Roatan, Honduras to explore these aspects. Experienced coral restoration volunteers did not perceive their contributions to coral health protection as being greater than control group non-volunteer divers, despite displaying a deeper understanding of the threats to coral reefs. Notably, both new volunteers prior to training and experienced restoration volunteers were more than twice as likely as control divers to highlight reducing carbon emissions as critical for coral health. While volunteer divers exhibit a strong baseline awareness and concern for coral reef threats, they report that participation in restoration efforts leads to significant learning gains regarding coral conservation. The insights gained from the motivations and learning outcomes of volunteers in coral restoration in Roatan may inform similar initiatives globally, potentially impacting a wide range of volunteer-based ecosystem restoration programs and diver-involved projects, thereby enhancing volunteer engagement and educational outcomes in environmental conservation efforts.

## Impact statement

Volunteer scuba divers are necessary for maintaining many coral reef restoration initiatives in the Caribbean. This article explores how a heterogeneous volunteer population in Roatan, Honduras perceives threats to coral reefs compared to control non-volunteer divers, as well as variation in their perceptions of restoration associated with gender, visitation status (residents vs. tourists) and other attributes. This research expands the question of how to determine restoration's success beyond ecological and economic metrics by examining how restoration participation can serve as a form of experiential environmental education for the non-experts who engage in it. This is especially the case for the restoration programs described in this research, which require all volunteers to complete classroom learning about coral restoration rationale and the state of the reef prior to the experiential component of volunteering. Volunteer-based coral restoration, in its best iteration, can empower already-concerned non-experts to engage within a community of like-minded divers. For visiting recreational divers in particular, volunteering for coral restoration may be the best or only opportunity they encounter to better understand coral reefs and the issues that threaten them. Understanding how distinct subsets of these volunteers view their work might help coral restoration practitioners leverage true ambassadorship for coral reefs. While the themes in this manuscript are most salient to Honduras's Bay Islands and the Western Caribbean, these findings may be extrapolated not only to coral restoration in other geographic regions but also potentially to other volunteer-based ecosystem restoration programs and other diver-engaged initiatives.

## Introduction

As coral restoration research and initiatives have proliferated in recent years (Bayraktarov et al., 2020; Boström-Einarsson et al., 2020; Hein et al., 2021; Shaver et al., 2022; Suggett et al., 2023), scholarship calling for more social research and success metrics into these reef restoration efforts has also grown (Anthony et al., 2020; Hein et al., 2017; Hughes et al., 2023; Gibbs et al., 2021; Le et al., 2022). In parallel, there has been increasing recognition of the potential of coral restoration initiatives to engage non-experts in protecting coral reefs (Hesley et al., 2023; Sebastian et al., 2024), as well as the mutual benefits of building collaborations between the dive industry, volunteer divers and restoration programs at scale (Howlett et al., 2022). Yet, despite the long-held assumption that increased volunteer knowledge and engagement for protecting coral reefs

will follow from engaging non-expert divers in coral reef restoration (Westoby et al., 2020; Suggett et al., 2023), and the explicit hope from some researchers that such engagement becomes a priority for restoration programs (Schmidt-Roach et al., 2020), there is little literature assessing baselines or outcomes of engagement, knowledge or perceptions of self-efficacy among these volunteers.

In their review of the published literature on motivations for coral reef restoration, Bayraktarov et al. (2019) found that "restoration for education purposes or awareness of the general public" as a motivation was considered "negligible"; nevertheless, many programs rely on subsets of the "general public" participation. Because reef restoration programs often employ both local and transient volunteers (Prideaux and Pabel, 2018; Lintangkawuryan et al., 2023), reef restoration efforts have potential to act as educational opportunities for non-experts (Hein et al., 2019; Escovar-Fadul et al., 2022; Helsey et al., 2023). Conversely, failing to impart an accurate understanding of the goals and limitations of restoration to volunteers and dive professionals could evoke unintended adverse consequences, from involved lay-people overestimating restoration as a long-term solution to coral reef decline to disillusionment with setbacks in restoration efforts (Fadli et al., 2012; Hein et al., 2019). Research on the overall effectiveness of education about coral reefs in tourism settings is also mixed, with some scholarship indicating its limited effectiveness in generating lasting attitude or behavior change (Hofman et al., 2020; Machado Toffolo et al., 2022), while others report significant increases in behavior-change intentions, pro-environmental attitudes and awareness (Powell and Ham, 2008; Machado Toffolo et al., 2022) in the short or long term.

Roatan, the largest and most-visited of Honduras's Bay Islands, provides a rich backdrop to investigate these topics. The Bay Islands are part of the Mesoamerican Barrier Reef System, which directly supports the livelihoods of more than a million people (Arrivillaga and Arreola, 2016). The pressures of rapid tourism growth in Roatan have presented numerous ecological and social challenges (Doiron and Weissenberger, 2014), with nearly a million annual short-term cruise ship visitors to Roatan alone accounting for half of Honduras's total tourist arrivals (Instituto Hondureño de Turismo, 2022).

In Roatan, several nonprofit organizations support tree-style shallow-reef coral nurseries (Nedimyer et al., 2011) with significant reliance on volunteer participation for routine operations and maintenance. The volunteers are a blend of tourists and residents, including locals and foreign nationals, who contribute hands-on support in maintaining and/or monitoring threatened *Acropora* spp. corals. Volunteers help clean the structures where the corals grow, learn to outplant coral fragments onto the reef and assist restoration professionals with coordinated outplanting efforts. As a precursor to volunteering with any of the coral restoration programs involved in this study, all divers are required to complete a training that includes a comprehensive overview of coral ecology, threats and restoration rationale and methods.

This research sought to understand these volunteers' perceptions of their contributions to protecting coral reef health, as well as their learning and affective engagement, across multiple similarly structured coral restoration projects in Roatan. Over 7 months, these coral restoration volunteers were surveyed, along with a group of comparable non-volunteering control divers, and interviewed to explore their understandings of how to protect coral reefs and the role of coral restoration. Data were analyzed using a mixed-methods approach, which include qualitative analysis of open-ended interviews and a semi-quantitative analysis of self-reported impact and knowledge. These findings suggest that participation in restoration efforts is highly self-selecting among divers already informed about coral reef threats, but nevertheless leads to significant learning gains regarding coral conservation.

## Methods

### Study site

All data were collected in Roatan, the largest of Honduras's Bay Islands off the north coast of the country. Roatan was the location of Honduras's first coral nursery, established in 2016 (Funderburk, 2023). The coral restoration programs involved in this research used the coral gardening technique in their in situ nurseries (Nedimyer et al., 2011) to propagate new colonies from fragments, and often partnered with various dive businesses to recruit and train volunteers.

### Data collection

All participants signed and dated digital consent forms prior to participation affirming that they understood the purpose and scope of the research (Supplementary Appendices 1 and 4). Moreover, this study adhered to the data sharing policy of the Coral Reef Alliance, which maintains the secure handling of personal identifiers and ensures that such information is not publicly disclosed or shared.

#### Surveys
Divers invited to complete the initial survey were either volunteers with prior coral restoration experience ($n = 21$), new restoration volunteers about to begin training via a course offered at an affiliated dive shop ($n = 38$) or control group divers ($n = 24$) at those same dive shops who were not participating and had never participated in restoration. Divers were incentivized to participate by the possibility of winning small prizes from a local NGO.

During the 7-month data collection period, all new participants in two coral restoration programs in Roatan, Honduras were invited to fill out a Google form before beginning their courses. The restoration course instructors were provided with a written paragraph inviting divers to contribute to research about "divers and coral reef health" when a researcher was not present to administer the survey. The surveys were offered in both English and Spanish (Supplementary Appendix 1).

This study defined experienced coral restoration divers as those who had already completed a training certification program for coral restoration prior to completing the initial survey. These experienced restoration divers were identified and invited to participate with periodic sharing of the survey link in a WhatsApp group used by one of the NGOs to coordinate routine restoration outings and by approaching dive professionals known to support restoration efforts.

Control group divers, who had never participated in coral restoration, were recruited to complete the survey in each of the five participating dive shops that offered a coral restoration specialty course through one of the restoration programs involved in this study. Each shop displayed recruitment posters with QR codes and bit.ly links to the survey. In addition, a researcher routinely went to each of these dive shops and asked divers to fill out the surveys; the vast majority of control group participants were recruited via this direct contact.

The surveys were pilot tested and refined with several local divers in January 2023 prior to data collection. They were administered via Google Forms and gathered the following information:

- Dive Experience: Lifetime dive experience was approximated by asking respondents to select whether they had completed 1–14, 15–59 or 60 or more dives. These intervals were chosen to reflect the minimum number of dives the NGOs require for divers to join the restoration course (15), and the minimum number of dives required to become a professional diver by the Professional Association of Diving Instructors standards (60).
- Bay Islands Dive Experience: Respondents were asked how many dives they'd completed in their lives in the Bay Islands with the same response options as the previous question.
- Prior Restoration Experience: Respondents who indicated that they had previously participated in coral restoration were asked when and where they had first participated, as well as how many total times they had done so.
- Restoration Course Taken: Whether the respondent had trained (or was about to train) in coral restoration through the Roatan Marine Park (RMP), Bay Islands Reef Restoration or a separate organization.
- Dive Shop: The certifying business that respondents primarily dove/trained with.

The surveys also posed three open-ended questions. The first two asked participants to list actions that a) individuals/communities and b) governments could take to protect coral reef health. The following explanation was provided: *For this survey, a healthy coral reef is defined as one with a high amount of living coral, large and diverse populations of marine life, and the ability to support human communities.* The third open-ended question was, "*In what ways, if any, do you intend to protect the health of coral reefs in the future? Please include anything you already do and intend to continue doing.*" Gender, age, place of origin and place of residence were collected at the end of the survey as optional demographic data. Participants could optionally indicate willingness to discuss their answers further in an interview.

Initial surveys (*n* = 83) were completed between January 25, 2023 and August 1, 2023. Surveys were submitted by more than three quarters of all new participants in one coral restoration program over the data collection period, and for a third of the new participants in another program over the same period.

### Semi-structured interviews

Semi-structured interviews were conducted with a subset of restoration learners and experienced restoration divers to gain a more robust understanding than would have been possible from surveys alone of a) how these divers perceived threats to coral reefs and their own contributions to protecting them, b) affective associations with coral reefs and restoration, and c) self-reported changes in knowledge or attitudes from restoration participation, positioned broadly in the interview questions as "takeaways." These questions were structured to elucidate cognitive, affective and self-reported behavioral components of participants' engagement with coral reef restoration.

The semi-structured interview questions and guidelines (Supplementary Appendix 2) were tested and refined for clarity with a newly certified restoration diver, an experienced restoration diver and a restoration instructor in January 2023. Prospective semi-structured interview participants were identified based on their willingness to participate as indicated in their survey and selected based on availability and willingness to proceed upon a researcher reaching out. For participants identified as restoration learners in the initial surveys, interviews were scheduled as soon as possible after the completion of their coral restoration training.

Fifteen semi-structured interviews were completed between January 25, 2023 and August 1,2023. Six of the interview participants were professional divers in Roatan at the time of the interview, including three dive instructors who taught coral restoration to incoming volunteers; the remaining eight were primarily recreational divers. For interviews conducted in English, initial transcripts were generated by Otter.ai, then reviewed for accuracy and lightly edited for clarity. Interviews in Spanish were transcribed manually, then translated to English with ChatGPT 3.5 and reviewed for accuracy by a bilingual researcher. After one interview was removed for poor audio quality, qualitative coding was performed on the remaining 14 semi-structured interview transcripts.

### Data analysis

All surveys and interviews were imported to Atlas.ti version 23.2.1 (Supplementary Appendix 3). Analysis was conducted and data were visualized both within Atlas.ti and in R version 2023.09.0+463.

#### Demographics
The number of survey respondents was visualized, grouped by their restoration experience and age, gender, place of origin, place of residence and scuba diving experience using the ggplot2 package (Wickham, 2016) in R.

#### Likert-type question
The Likert-type survey responses asking participants to rate their agreement with "I do a lot to protect the health of coral reefs," on a scale from "Strongly Disagree" to "Strongly Agree" were assigned numeric equivalent responses (1 = "Strongly Disagree" and 6 = "Strongly Agree"). The values were then fitted to an ordered logistic regression model with the "polr" function in R's MASS package (Venables and Ripley, 2002) with variables including coral restoration experience, experience diving in the Bay Islands, gender and place of origin. The "stepAIC" function was used for stepwise selection to determine the optimal model, then tested for goodness of fit by running a type II ANOVA with Wald chi-square tests on a cumulative link model using the "clm" function in R's ordinal package (Christensen, 2023) with the selected parameters.

#### Open-ended survey questions
The qualitative coding of open-ended survey responses and semi-structured interviews was performed in Atlas.ti version 23.2.1. Responses to the three open-ended survey questions about ways to protect coral reef health were coded in several iterative rounds. "Don't know" and "nothing" were each coded as well; answers that did not emerge among multiple respondents were coded as "other."

These data were treated as binary in analysis, for example, 0 to indicate absence of a particular code for a given respondent or 1 to indicate the code's presence. Because of the relatively small sample size, Fisher's Exact Test ("fisher.bintest" in R package RVAideMemoire) was used instead of a chi-square test to determine the significance of the relationships between certain variables and responses, including how restoration experience and gender correlated with the likelihood of mentioning lowering carbon

emissions or coral restoration as a way to protect coral reef health. The codes representing the 10 most common responses to the open-ended questions were visualized using ggplot2 in R, grouped by the three restoration experience groups analyzed (control, learner and experienced).

### Interviews

Interview transcripts were coded inductively in several iterative and reflexive rounds of thematic analysis (Guest et al., 2011) within Atlas.ti to identify and refine recurrent emergent themes. A blended deductive–inductive approach (as described by Proudfoot, 2023) was used for coding, with interviews read closely for any emergent themes, as well as for themes elucidating participants' cognitive, affective and behavioral engagement with coral reefs and restoration.

## Results

### Demographics

All demographic data collected are displayed in Figure 1. Survey respondents were predominantly from the United States ($n = 49$), with Canada ($n = 14$), Europe ($n = 10$) and Honduras ($n = 6$) being the next most common places of origin. Fewer than 11% ($n = 9$) of survey respondents were from Latin America. The experienced restoration volunteers skewed female when compared with both the control group and the restoration learners (76% of experienced restoration volunteers surveyed were female, whereas restoration learners and control divers were evenly split).

Experienced coral restoration divers also had more overall dive experience as well as experience diving in the Bay Islands, while control group divers were less likely to have experience scuba diving, overall or in the Bay Islands.

### Surveys

#### Likert-type model selection

The model that minimized the Akaike information criterion (AIC) value for the Likert-type question responses (Figure 2) included both coral restoration experience and Bay Islands diving experience as positively correlated with higher rankings on the Likert-type question, but excluded gender and place of origin (AIC = 248.72). The goodness of fit test returned a significant result for coral restoration experience ($p = 0.01616$) and a marginally significant result for participants' number of dives in the Bay Islands ($p = 0.06052$). The model was very significant for control divers compared with restoration learners ($p = 0.00882$) and marginally significant for restoration learners compared with experienced restoration divers ($p = 0.05097$), but was not significant for the control group compared with experienced restoration divers ($p = 0.72093$). It was also significant for the lowest level of Bay Islands dive experience compared with both the intermediate ($p = 0.04295$) and highest ($p = 0.04158$) tiers of participants' total Bay Islands dives.

#### Open-ended questions

Open-ended survey responses about protecting coral reef health were coded as 632 distinct segments, with a median of two coded segments per open-ended question per respondent. Recurring responses about protecting coral reef health were ultimately grouped into 12 distinct codes (Table 1).

Divers at all levels of restoration experience (control divers, restoration learners and experienced coral restoration divers) were similarly likely to list coral restoration as a way to protect coral reef health for at least one of the open-ended questions about protecting coral reef health. However, the frequency of other responses varied notably by restoration experience, for example, 18% of control divers writing at least one response coded as

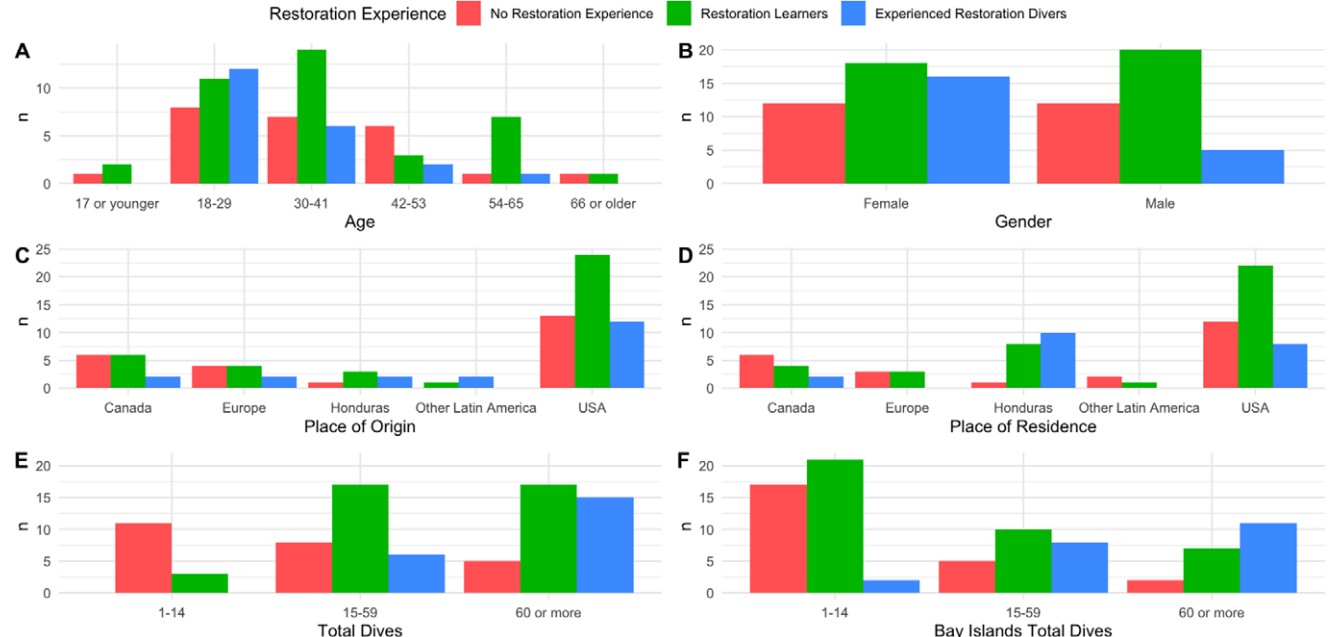

**Figure 1.** Number of survey respondents grouped by their coral restoration experience and plotted by (A) age, (B) gender, (C) place of origin, (D) place of residence, (E) overall diving experience and (F) Bay Islands diving experience. Not all plots total to 83 (the number of surveys collected) due to the exclusion of less common answers (e.g., $n = 1$ for "India" as place of origin).

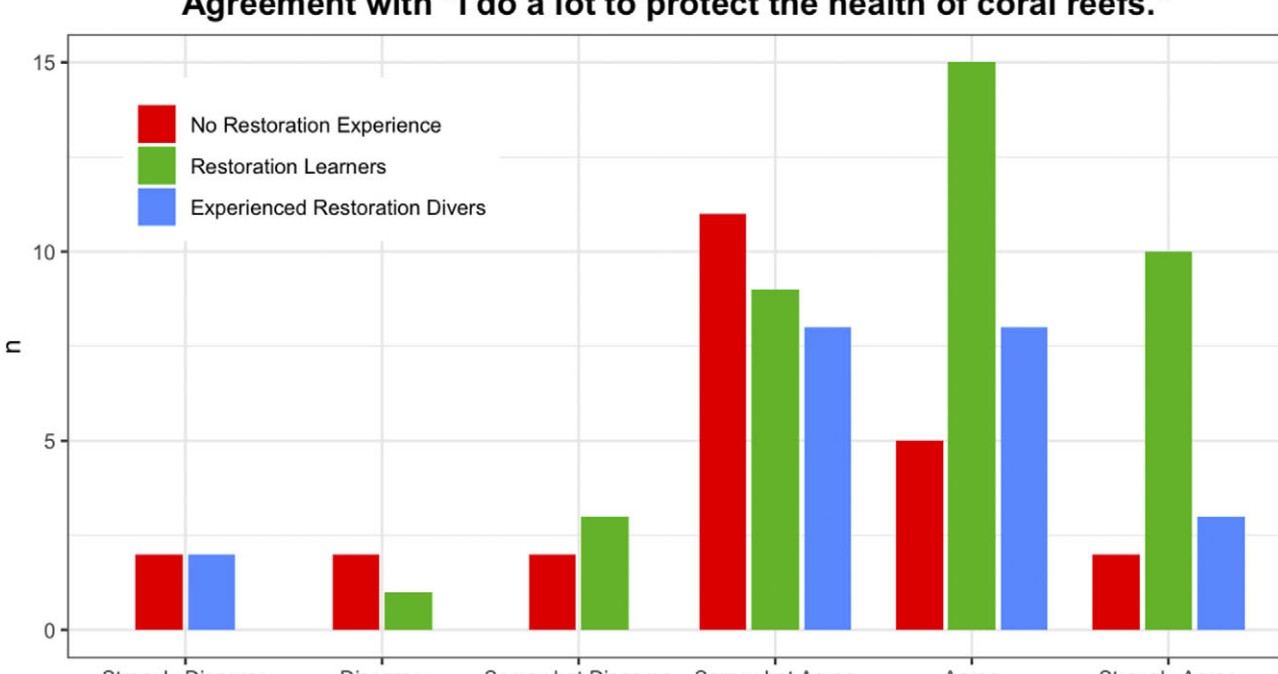

**Figure 2.** Survey responses (*n* = 83) expressing level of agreement with the statement "I do a lot to protect the health of coral reefs," grouped by level of restoration experience.

fisheries-related compared with 44/45% for experienced restoration divers and restoration learners, respectively (*p* = 0.04396, Fisher's exact test).

The "lower carbon emissions" response showed a similar contrast, with fewer than 14% of control group divers identifying "lower carbon emissions" as a way to protect coral reef health compared with 32% of restoration learners and 30% of experienced restoration divers; however, this difference was not statistically significant (*p* = 0.2006, Fisher's exact test). There was a significant difference (*p* = 0.0236) based on gender, with 14% of male participants listing lower carbon emissions in at least one of their open-ended responses compared with 37% of females.

The top 10 most common responses overall are shown with the proportion of survey respondents from each tier of restoration experience who gave that response in Figure 3.

### Semi-structured interviews

A blended inductive and deductive approach yielded a median of 36 coded segments per semi-structured interview, with themes related to divers' motivations, perceptions and behavior (Table 2). Participants most commonly described coral restoration in terms of its experiential or "hands-on" qualities, as well as in reference to the scope and/or scale of the work (each brought up in 10 of the 14 interviews, or 71%). Coral restoration was also described as challenging (five interviews/36%), fun or enjoyable (four interviews/29%) and relaxing (three interviews/21%).

Both newly certified restoration divers and seasoned professionals often discussed experts and science, typically in the context of minimizing their own experiences or contributions. Participant 2, a recreational diver who had recently completed restoration training and decided to begin professional dive training at the time

of the interview, commented in regard to her self-appraisal of her own efforts to protect coral reef health:

I'm not a marine biologist. I'm not a professional. I'm not someone who's doing anything in that sense. I am purely a recreational diver – well now, hooray, a professional diver. But I'm not a scientist, if you know what I mean…I agree that I'm doing something, and previously as a purely recreational diver, I was actively getting involved. But yeah, you certainly can't be the person who says 'I strongly agree that I am doing everything I can for the reefs', because you know, that's not my job.

Participant 18, a dive professional who teaches and certifies new coral restoration volunteers, expressed similar reservations:

I am working on it quite a bit but at the same time, it's not completely because I am not a scientist. I am not a marine biologist. That is not my background. My background is a dive professional…

Participant 18 went on to describe an extensive career of marine stewardship and advocacy. Other participants emphasized experts and scientists as a source of hope for coral reefs because of their work in coral restoration, but positioned themselves outside of contributing to that work. Participant 21, a short-term visitor interviewed after completing the restoration training, encapsulated these perspectives:

I think it's really, really cool to see people who are dedicating their lives to that. I think that's the thing that probably sparked some hopefulness is seeing the dedication that you guys have. It makes me – it makes me want to do it. I'm a photographer, I'm not a scientist, so I won't….

Other emergent themes included doubt or uncertainty, including personal doubt (e.g., participant 63, "You're not sure if you're helping"); restoration process doubt (e.g., participant 76, "It also made me feel the slightest bit like 'okay, this algae is just gonna grow back'…hopefully this helps the coral because I'm being told that it does, so I'm going to do this, but part of me is like, I don't

**Table 1.** Top responses coded in response to the three open-ended survey questions: *List any actions you think* **individuals** *or* **communities** *could take that would help protect the health of coral reefs*; *List any actions you think* **governments** *could take that would help protect the health of coral reefs*; and *In what ways, if any, have* **you** *helped protect the health of coral reefs in the last year?*

| Reef health subcodes | Description | Example |
|---|---|---|
| Sunscreen/personal products | Includes "skincare products" and personal hygiene products, or references to "reef–friendly" or "reef–safe" products. | "We use the proper insect spray and sunscreen to minimize harmful chemicals in the water." |
| Responsible diver and recreator behavior | Diver, snorkeler and swimmer behaviors. Commonly identified in–water behaviors include buoyancy, dive control skills and individual–level interactions with marine life. Also includes actions of individuals recreating around the reef (e.g., boating and fishing for sport) and actions of recreational operators (e.g., boat anchors and traffic). | "Advocating safe reef practices to other divers and individuals, such as…good dive habits, particularly buoyancy. A strong emphasis on good buoyancy on all dives for all divers." |
| Reduce plastic/litter | Including cleanups of existing litter and source reduction, waste management and recycling. | "Making marine debris reduction easier, e.g., plastic reduction schemes and proper refuse facilities." |
| Coral restoration | Includes reef restoration, coral nursery and outplanting. | "Participate in coral restoration at some point in the future." |
| Education | Includes learning, kids/school, teaching friends/peers and variations of educating "locals" or "local communities." Also includes "awareness." | "Education for kids in school, and adults who work in and around the reef, whose jobs impact the health of the reef, with incentives, penalties… and include tourists." |
| Fisheries | Including seafood choices, broadly supporting healthy fish stocks and explicit mentions of Marine Protected Areas (MPAs); excludes fishing line and tackle. | "Create strict policies that make it more difficult to illegally fish and harm these ocean ecosystems." |
| Support institutions | Includes supporting science, conservation, governments programs and scholastic organizations. Examples: funding, donating, research, capacity–building and volunteering. | "Invest in research on coral health." "Encourage the community to participate more in Roatan Marine Park projects." |
| Lower carbon emissions | Includes mentions of climate change, global warming, carbon footprint, greenhouse gasses, ocean acidification and/or related terms and concepts. | "On a broader climate scale, being conscious of my individual GHG impacts and, through my job, working to reduce atmospheric GHG's through US federal policy." |
| Reduce pollution | Includes broad mentions of "chemicals." This category was used when the response mentioned pollution without specifying what kind and/or in a manner that did not clearly fall solely under "Reduce land–based threats" or "Reduce plastic/litter." | "Controlling the use of toxic products that can end up in the sea in communities near it." "Control runoff, rainy season, rivers, to avoid pollutants." "Cruise ships stop dumping." |
| Reduce land–based threats | Includes coastal development, land–based runoff and protecting coastal habitat like mangroves. | "Controlling the use of toxic products that can end up in the sea in communities near it." "Implementing stricter coastal construction regulations on island, protecting mangrove forests." |
| Invasive species | Includes mentions of lionfish, a widely recognized invasive Caribbean species. | "Kill organisms that don't originally belong to the coral reef and are affecting the habitat." |
| Protected areas | Includes MPAs, suggestions of limiting or prohibiting various uses of the ocean. | "Creating more protected areas and enforcing regulations, conducting effective patrols." |

know.") and moral doubt (e.g., participant 69, "Sometimes I wonder if it's okay for humans to intervene so much in these things…it's like the manipulation of all this, right? But then I say, 'Okay, humans have put their hands in many things, sometimes for good, sometimes for bad'. So I just think, 'I hope this is for the good'. Because nature itself knows and regenerates itself, too."). At least one expression of doubt or uncertainty occurred in half of the interviews.

Scuba diving culture and tourist behavior were also recurrent emergent themes, with at least one of these discussed in 79% of the interviews. Several dive professionals emphasized the importance of serving as role models and safeguards against destructive practices (e.g., participant 47, "We [dive professionals] also have to take care of our part. When we have visitors, maybe they are not aware, but we are."). Tourists placed responsibility on other tourists more so than on dive professionals (e.g., participant 49, a short-term visitor, who said, "All their sunscreen, all them with no consciousness about [how] you can't touch the reef, you can't step on the reef, you need to protect it … There's just so many

things that people are not educated [about]… it makes me so sad and scared for that impact, the cruise ship impact, and the people who just come off the ship.").

## Discussion

Recreational and professional scuba divers can be valuable allies for ocean conservation goals (Forrest et al., 2023), including coral restoration efforts (Hesley et al., 2023). Through surveys and interviews conducted with divers in Roatan, Honduras, this research provides nuanced insights into the attitudes and perceptions of divers engaged in coral restoration. The study reveals that divers with experience in coral restoration programs do not necessarily perceive their contributions to coral health as surpassing those of non-participating divers (Figure 2), even though they demonstrate a more profound understanding of coral reef threats (Figure 3). Volunteers involved in coral restoration were more than twice as likely as their non-participating counterparts to note the

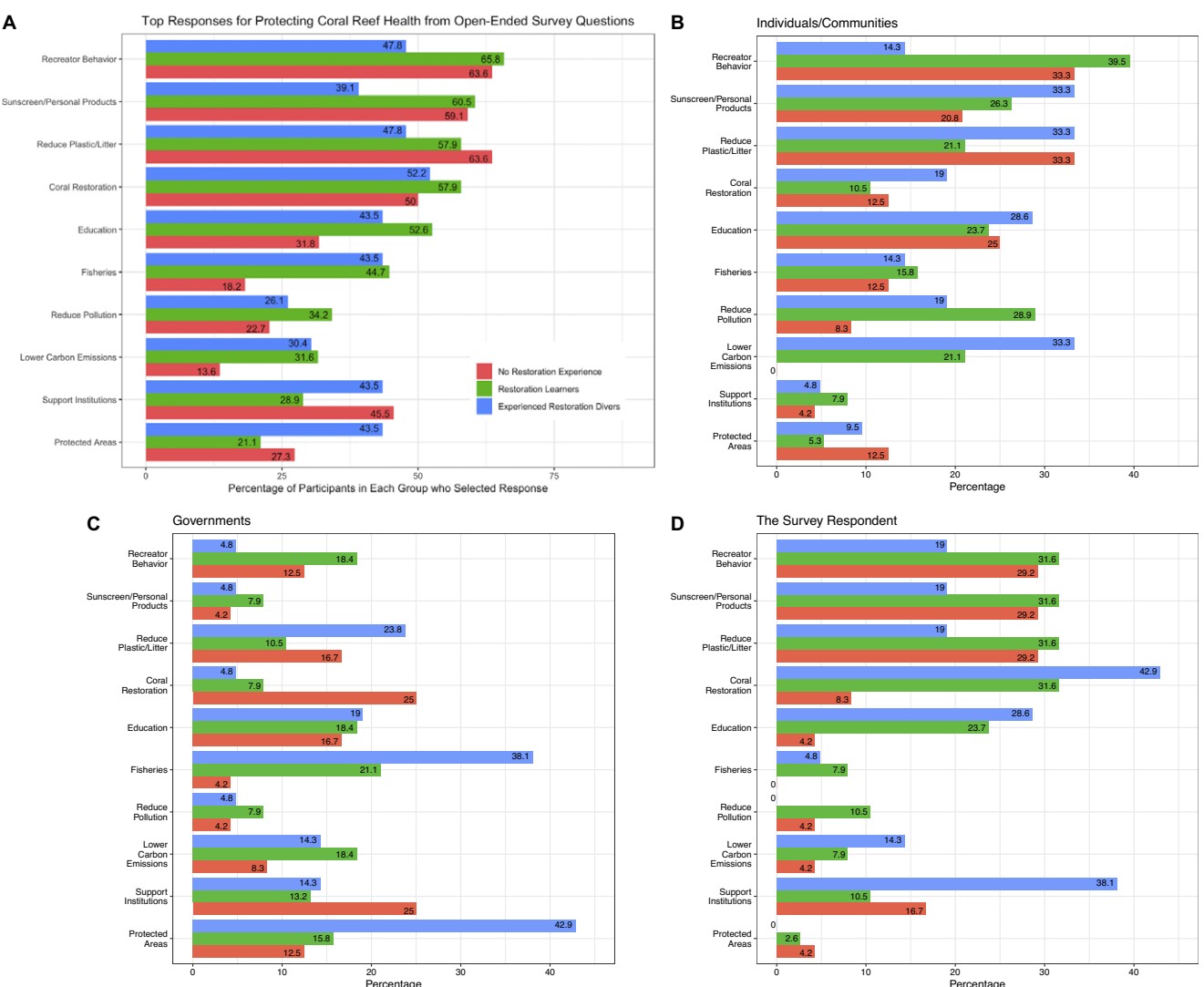

**Figure 3.** Top 10 ways to protect the health of coral reefs identified by participants in response to (A) any of the three open-ended survey questions, (B) individual or community actions, (C) government actions, and (D) personal actions (those the participant personally takes or intends to take). Responses were aggregated with survey questions, "List any actions you think **individuals** or **communities** could take that would help protect the health of coral reefs"; "List any actions you think **governments** could take that would help protect the health of coral reef" and "In what ways, if any, have you helped protect the health of coral reefs in the last year?" Responses are ordered by the proportion of restoration learners (green bar) that indicated a given response for at least one of the three open-ended survey questions. See Table 1 for the descriptions and examples of the open-ended responses.

importance of reducing carbon emissions for coral health. However, they are not more likely than control divers to identify coral restoration as a way to protect reef health. This study builds upon the premise of Sebastian et al. (2024) that "community-based restoration efforts can be utilized as a conservation education tool to promote ecological awareness, stewardship and inspire future conservationists." While volunteers display a strong baseline awareness and concern for coral reef threats, their participation in restoration projects nevertheless enhances their understanding of coral conservation.

### Awareness of threats to coral reefs

These findings suggest some support for the popular concept among reef restoration practitioners that engaging in coral restoration can raise engagement with coral reef threats (Schmidt-Roach et al., 2020). However, the survey responses and interviews more

clearly indicate that participation in coral reef restoration is a self-selecting activity among divers already concerned with the well-being of coral reefs. This is particularly evidenced by the fact that when restoration learners enroll as volunteers, they demonstrate a stronger baseline awareness than control divers of the need to lower carbon emissions (Kleypas et al., 2021; Hoegh-Guldberg and Manzello, 2023) to protect coral reefs (Figure 3).

Conversely, control divers were the most likely group to offer responses about protecting coral reef health in the "reduce plastic/litter" category (Figure 3), although this response was among the most common for all restoration groups. This supports prior research (e.g., Dean et al., 2021; Apps et al., 2023; Waters et al., 2023), suggesting that divers and general non-experts have a mixed understanding of the major threats facing coral reefs and may focus on visible threats like plastic or diver behavior (Dearden et al., 2007) while underestimating the importance of key actions for coral reef survival, such as curtailing greenhouse gas emissions.

**Table 2.** Themes coded in at least 20% (three or more) of semi-structured interviews

| Theme | Sub-theme | Prevalence | Example(s) |
|---|---|---|---|
| Motivation for restoration involvement | Better Future | 21% (n = 3) | I hope for a future where I can continue to enjoy them and my children can enjoy them. |
| | Economic | 21% (n = 3) | That's our responsibility because, you could say, we live for this. |
| | Enjoyment | 36% (n = 5) | One of my biggest takeaways, like if someone asked me what is the reason of why I like doing it, you can see the beauty which are the colors and structures, they are just amazing and it blows you away and that's what motivates me to keep on doing it. |
| | Learning | 29% (n = 4) | I love the ocean. I think it's so interesting, but I honestly feel like I know very surface level things about it. Like always wanting to learn as much as I can. Because if I'm kind of an ambassador for who I work with and what I represent, I want to be educated in the field, you know. |
| | Not Doing Enough | 43% (n = 6) | It's exhausting to constantly try and … we cannot, I cannot change everything. I can just change me. And so I try to, I try to embody that and just acceptance of like, well, I can only – this is where I – this is my zone of influence, basically. |
| | Purposeful Contribution | 43% (n = 6) | And it felt nice to actually be contributing instead of just admiring the fish and the reef and stuff but trying to contribute to helping out. |
| | Sense of Responsibility | 29% (n = 4) | I think somebody have to give back. |
| Restoration experience | Challenging | 36% (n = 5) | It's hard work. No, I never thought it was so – well, it's not exhausting, but it's very meticulous and a lot of process. One might think it's just like putting the coral there and seeing what it does, but really, all the control over the coral, checking its quality, whether it's sick or not, the fragmentation part, and putting another coral back, the cleaning. Everything was a fairly demanding job that people who are not involved are not aware of what is really done down there. |
| | Experiential | 71% (n = 10) | …it was like the same as doing something on land, just underwater, like gardening. You're gardening native plants and you are pulling weeds out. It's like the same thing but underwater. I was kind of reflecting on – I think often stuff in the ocean feels like more of a like big leap or big – it's a much different – you are diving, but it wasn't hard. It was fun. |
| | Fun/Enjoyable | 29% (n = 4) | I really enjoy cleaning in the nursery when we are cleaning. That's fun, and then when we do outplanting, looking for the best places to do the outplanting and everything, I like that too. |
| | Relaxing | 21% (n = 3) | I think it was very just like meditative. I enjoyed the experience of just kind of being very single focused. When I found out for the second dive that we'd been out there for like, 70 min, I was like, "What? No way, that can't be right." So time passed quickly. Yeah, and it was nice to have a simple task that felt manageable and easy to understand, easy to do. |
| | Effort, scale and scope | 71% (n = 10) | "Oh wow, it's a lot of effort and a lot of work that takes a lot of manpower." So I was like, "this is cool, this feels productive. I feel like I'm helping right now." I definitely felt that but I think we need a lot more people, a lot more of these nurseries… the scale does not feel big enough to make the impact that I imagine needs to be made. |
| | Technology | 21% (n = 3) | I've always felt like it's very low tech…It reminds me a lot of the agricultural industry which I came from, and we would do like crossbreeding, but it was like very, it was very, very low tech, like you'd literally collect pollen, and then like use the tip of an eraser to put that pollen on to like the stamen of a flower … And this is very similar to me. So it's very labor intensive, very kind of like low tech. But it's impactful, I guess. |
| Perceptions of groups connected to coral restoration | Community | 63% (n = 9) | People take away a lot of knowledge about the reef, and we also try to awaken the same feeling we have towards it because once they have that passion, they start to inform themselves more, they start telling others, it's a chain. |
| | Divers | 71% (n = 10) | If you are a really good diver who never touches coral but you go through three plastic bottles a day that has an effect. And you know, if you are a great diver but actually if you are sort of using some loads of sunscreen that is not reef safe and this that and the other and then you go diving, that has an effect. And anything that you do that it leaves a carbon footprint. It has an effect on what happens in our oceans, you know? And yeah, I sort of feel like it would be a good time to start extending that reach of what we are trying to tell people to do, because it's very much in the media and in people's minds at the moment. |
| | Experts/Science | 71% (n = 10) | I'm not a scientist and I can come up with kind of creative solutions to the problems in front of me, but I want genius 20 year olds to study coral reefs enough to understand how to actually fix things, in a way that I feel like I'm incapable of with my background. |
| | Tourists | 50% (n = 7) | I think the main one, the biggest one would be to be a mindful tourist and to educate yourself further. I think those are the two if everybody just can manage to do that to just be conscious when they travel. And to be – and to try and educate themselves, or read some articles about it, or whatever it is to further educate themselves a little bit about you know, coral reefs and their threats. Then that would already make a huge difference. |

**Table 2.** (*Continued*)

| Theme | Sub-theme | Prevalence | Example(s) |
|---|---|---|---|
| | What others know | 50% (*n* = 7) | So it's just awareness, all of this education, many, many, many people for us, it's obvious because we are divers, right? But many, many when I go home, you know, this is like news to my friends, because they do not they do not know they have never been in the environment. So I think that that's – we kind of assume working in the business that everybody's already educated on this and that is far, far from the truth. |
| Learning/knowledge | Coral Ecology/ Physiology | 57% (*n* = 8) | I had a vague idea that you know, coral was an organism not like just a plant or a rock sort of thing, but we went into a lot more detail about that and talking about – and I still now it happened only a couple of months ago, and I'm going, "I need to go do a bit more reading on that," because, you know, they told us a lot about the polyps and the way that it happens and the things that can actually affect that and, you know, bring down the bring down the lifecycle of the polyps themselves. And actually yeah, I do not know if I've ever really thought about it before." |
| | Reef Threats and Solutions | 57% (*n* = 8) | I suppose if you asked me two months earlier, I would have been like, "uhhhh, pollution." But…we all know that global warming – well, I'm saying that – we should all be aware that global warming is a thing. And you know, the small changes in temperature in the ocean, even by one degree is having an effect on our coral reefs. So, it's more than just people saying, "Oh, well, I won't touch the corals when I'm in the water." I think people need to understand that if you really want to be proactive about your coral. It's everything. It's what you are doing on land that's going to affect what happens in the sea as well. |
| | Restoration Process | 43% (*n* = 6) | And planting, coral planting I thought was really interesting. That's not how I would have expected that it happened. I do not know what I was expecting, but I thought that was really interesting. |
| Changes from restoration participation | | 86% (*n* = 12) | Maybe it was a bit…idealized in my brain when I had just started off, thinking like "Oh!", you know, "we are gonna make coral reefs live again." And, I mean, obviously, it's not that simple. |
| Sharing with others | | 93% (*n* = 13) | And then we also, I think we also just had some good discussions in our group, chatting about like, what does this mean? What does this look like, is there – where's the hope? |
| Feedback | | 63% (*n* = 9) | How are we supporting people in their daily lives, to make that possible financially for those people to take time? |
| Individual action | | 50% (*n* = 7) | This is my first time ever getting involved with coral restoration, so it felt like something I was excited to learn more about and take action in an individual way. I think that generally, the most effective things are going to come on like big scale projects and like government funded or like, you know, people really coming together at a systemic level. |
| State of the reef | | 93% (*n* = 13) | Since I came to live on the island, many people in Tegucigalpa, where I'm from, ask me, and people here in Honduras too, and I try to share and tell them, "the reefs are sick, they need our help, try to be aware." |
| Doubt/uncertainty | | 50% (*n* = 7) | So part of me feels like, am I – how much does one piece of coral really matter? |

### Gender and coral reef threats

A notable gender gap emerged in the responses about protecting coral reef health, with 37% of all female respondents providing a response in the "lower carbon emissions" grouping for at least one of the questions compared with 14% of men. Of the five males in the experienced restoration diver category, none gave responses that included "lower carbon emissions," and women were more likely to bring up lowering carbon emissions than their male counterparts in every restoration experience group. This difference is particularly striking since lowering carbon emissions is considered by most experts to be the chief requirement for coral reef survival (Kleypas et al., 2021; Hoegh-Guldberg and Manzello, 2023).

The gender gap in identifying carbon emissions as necessary for protecting coral reef health may reflect the well-documented tendency for women across demographic groups to have more knowledge and concern on average than men about both climate change and broader environmental issues (McCright, 2010; McCright and Xiao, 2014). Although there is some debate as to how universal the climate and environment gender gap is outside Western nations (Knight, 2019), the high proportion of U.S. respondents in this study (59%) makes a stark gender divide plausible.

### Perceptions of expertise

Experienced coral restoration volunteers demonstrated a more comprehensive understanding of coral reef threats than control group divers, but did not assess themselves significantly differently from control divers in agreement with "I do a lot to protect coral reef health" (Figure 2). Several interviews further support the notion that restoration divers, whether new learners or experienced, are hesitant to characterize their own contributions to conserving the reef as meaningful compared with the efforts of scientists or experts, who they perceive as more impactful. Interview subjects who acknowledged that they knew more, cared more or acted more than a typical diver on behalf of coral reefs were then quick to discount these comparative strengths.

One possible interpretation aligns with the tendency of competent people to underestimate their own abilities (Kruger and Dunning, 1999), hypothesized to be in part due to a better metacognitive assessment of "knowing what they don't know" compared with less competent populations but also due to high-performers overestimating others' competence (Dunning, 2011). Highly motivated and experienced restoration divers may therefore more accurately recognize the complexity and challenges of protecting coral

reef health than their less experienced counterparts, comparable to Fuchs's (2023) finding that "attitude and intention deteriorated" with gains in environmental education among college students. The comparative nature of volunteers' low self-assessments of their contributions, for example, by invoking scientists, researchers or professionals in coral conservation, may merit further research and consideration in shaping communities of restoration involving non-experts, as the authors found little literature documenting or refuting this phenomenon.

### Limitations

An opportunistic subset of the control population (anyone who dove at the same dive shops administering the restoration courses during the time period when the study was conducted) was asked to complete the survey. Although any eligible diver could contribute by scanning the QR codes in participating dive shops, in practice nearly all control surveys were acquired by direct interaction when approached by a researcher after or between dives. This may have resulted in unintentional sampling bias in control group divers. The survey format may also have biased control group divers toward considering coral restoration in their open-ended responses, since all survey respondents were required to select their level of coral restoration experience before answering the other survey questions.

In addition, demographic data did not distinguish between Hondurans from the Bay Islands and those originally from the Honduran mainland, a potentially relevant oversight since many Bay Islanders have substantial linguistic and cultural differences from mainland Hondurans (Stonich et al., 2009) that may uniquely influence their relationships to the reef. However, since a small proportion of respondents were originally from Honduras, the authors do not believe this significantly biased the results.

Finally, it is relevant to note that the scope of this research focused only on coral restoration programs run by NGOs with reef conservation as core to their stated missions, and which explicitly teach restoration volunteers why specific restoration choices had been made and why the organizations were undertaking coral restoration in the first place. The NGOs involved in this research also mandate that volunteer divers learn about the wide range of threats to Roatan's reef system before they begin learning specific restoration techniques; explanations of the historic and present-day causes of reef decline take 30 minutes or more. However, other coral restoration initiatives across the world may be far less thorough in informing volunteer divers. In addition, other coral restoration efforts can arise from objectives such as mandated remediation for environmental damage or for-profit entities (Suggett et al., 2023), which may intend coral restoration to bolster their public image. This distinction matters in particular for research on engagement outcomes for volunteers, both in terms of the baseline engagement and knowledge of people who self-select to participate and in what they take away from the experience.

### Conclusions and recommendations

This study underscores the significant role that coral restoration initiatives play in elevating awareness and deepening understanding of the critical threats facing coral reefs. Participants in these programs exhibit a higher baseline commitment to reef protection, driven by their direct involvement and increased awareness of environmental challenges. This engagement highlights coral restoration not just as a conservation strategy but as an effective educational tool that enhances ecological consciousness among community members and volunteers.

However, our findings also reveal a noteworthy communication challenge within the coral restoration community. Despite their active participation and substantial contributions to reef health, many volunteers grapple with a sense of doubt and a perceived lack of legitimacy in their efforts, primarily because they do not identify as scientists. This sentiment reflects a broader misconception that scientific expertise is the sole custodian of environmental stewardship.

Addressing this issue presents a critical opportunity for coral restoration programs. There is a pressing need to broaden the narrative to include all stakeholders in environmental conservation efforts. By emphasizing that every individual's efforts are valid and valuable, restoration programs can empower participants and reinforce the idea that the responsibility for our planet's health is shared, not exclusive to the scientific community. Ultimately, expanding this inclusive approach will not only enhance the effectiveness of coral restoration efforts but also foster a more engaged, informed and committed global community dedicated to sustaining the health of our oceans.

Further research is necessary to evaluate whether the findings outlined here can be applied in broader contexts, such as coral restoration programs with similar volunteer models outside of Roatan. Related research questions, such as which educational interventions might effectively boost volunteer engagement with non-restoration coral health measures, and how to best empower participants to view themselves as meaningful contributors and ambassadors, are ripe for future exploration.

**Supplementary material.** The supplementary material for this article can be found at http://doi.org/10.1017/cft.2024.13.

**Open peer review.** To view the open peer review materials for this article, please visit http://doi.org/10.1017/cft.2024.13.

**Data availability statement.** All data and detailed explanations of how analysis was conducted on the data are available in the Cambridge Open Engage repository at https://www.cambridge.org/engage/coe/article-details/658dee d6e9ebbb4db9f0a2b0.

**Acknowledgements.** This research would not have been possible without support from partner organizations including the Roatan Marine Park and Bay Islands Reef Restoration, as well as the funding and support of the Fulbright Program and National Geographic Society. Yolanda Lee Waters contributed valuable early feedback on the research design. Juan Manuel Castro Antunez and Verónica Coates back-translated the survey and semi-structured interview questions, respectively, from Spanish to English to independently test consistency between the versions. Aditya Khandelwal advised on code in R.

**Author contribution.** Conceptualization: S.G.; Data curation: S.G.; Formal analysis: S.G.; Investigation: S.G.; Methodology: S.G., A.R.; Original draft: S.G.; Review and editing: A.R.; Software resources: A.R.; Supervision: A.R.; Validation: A.R.

**Financial support.** Funding for this research came from the U.S. State Department Bureau of Educational and Cultural Affairs Fulbright Program as a 2022–2023 Fulbright-National Geographic Storytelling award.

**Competing interest.** The authors declare no competing interests.

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
