## [Editor Report]

Both reviewers conclude that this topic is timely and relevant. Reviewer 1 points out some major structural issues that should be comprehensively addressed in order to consider the manuscript for publication. In particular, the authors need to:

- Be more explicit about survey design and methods for coding, 

- Move details of data analysis from the Results to the Methods section,

- Move specific descriptions of the outcomes of the data analyses from the Discussion to the Results section,

- Use the Discussion section to provide a more comprehensive overview about how the results obtained from this study are consistent with or diverge from the broader literature around coral reef restoration and engaging citizen science in management practice, and

- Rename the Additional Discussion section to something like “Research Limitations”

If these major structural changes are made, I don’t believe there would be any need for a Conclusions section.

Lastly, this is a study involving Human Subjects Research, which would typically require ethics approval. Please provide information for any ethics approvals obtained.

---

## [Editor Report]

In addition to the minor areas for revision raised by Reviewer 1, below are a few other minor points:

p. 7, line 41: I believe this should be "code’s"

p. 14, lines 3-7: Suggest editing the participants response so that they don’t come off sounding less intelligent. It is a kind thing to do. You are not changing their content, but just taking out the parts of verbal speech that when written down come off poorly. Suggested revision:

"I think it’s really, really cool to see people who are dedicating their lives to that and it’s for me . . . the thing that probably sparked some hopefulness is seeing the dedication that you guys have. It makes me—it makes me want to do it. . . . I’m a photographer, I’m not a scientist, . . ."

p. 16, line 23: This should be "comparable to Fuchs (2023)’s finding"

Lastly, the authors' response regarding the nature of the research falling clearly within the scope of low-risk human studies exempt from IRB oversight under Common Rule is probably applicable for this seemingly very low-risk questionnaire. However, moving forward, if the intent is to conduct further human subjects research, I would urge the Coral Reef Alliance to consider developing its own internal IRB process or partnering with an institution who does have an IRB. Just because the surveys were reviewed by the Bay Islands National Marine Sanctuary Technical Committee does not necessarily ensure that anyone on that committee has received the proper training to make assessments of risk in research as it pertains to human subjects. Moreover, some journals will not publish social research that has not had full ethics or IRB approval.

---

## [Editor Report]

The authors have adequately responded to the further revisions requested. This is acceptable for publication.